# Assessment of Sound Source Tracking Using Multiple Drones Equipped with Multiple Microphone Arrays

**DOI:** 10.3390/ijerph18179039

**Published:** 2021-08-27

**Authors:** Taiki Yamada, Katsutoshi Itoyama, Kenji Nishida, Kazuhiro Nakadai

**Affiliations:** 1Department of Systems and Control Engineering, School of Engineering, Tokyo Institute of Technology, 2-12-1 Ookayama, Meguro-ku, Tokyo 152-8552, Japan; itoyama@ra.sc.e.titech.ac.jp (K.I.); nishida@ra.sc.e.titech.ac.jp (K.N.); nakadai@ra.sc.e.titech.ac.jp (K.N.); 2Honda Research Institute Japan Co., Ltd., 8-1 Honcho, Wako, Saitama 351-0188, Japan

**Keywords:** drone audition, sound source detection, sound source tracking, microphone array

## Abstract

Drone audition techniques are helpful for listening to target sound sources from the sky, which can be used for human searching tasks in disaster sites. Among many techniques required for drone audition, sound source tracking is an essential technique, and thus several tracking methods have been proposed. Authors have also proposed a sound source tracking method that utilizes multiple microphone arrays to obtain the likelihood distribution of the sound source locations. These methods have been demonstrated in benchmark experiments. However, the performance against various sound sources with different distances and signal-to-noise ratios (SNRs) has been less evaluated. Since drone audition often needs to listen to distant sound sources and the input acoustic signal generally has a low SNR due to drone noise, making a performance assessment against source distance and SNR is essential. Therefore, this paper presents a concrete evaluation of sound source tracking methods using numerical simulation, focusing on various source distances and SNRs. The simulated results captured how the tracking performance will change when the sound source distance and SNR change. The proposed approach based on location distribution estimation tended to be more robust against distance increase, while existing approaches based on directional estimation tended to be more robust against decreasing SNR.

## 1. Introduction

Drones have recently been required to be used at disaster sites. By attaching auditory sensors, drones can obtain auditory information, and this field of auditory information processing for drones is referred to as “drone audition”. Drone audition techniques enable drones to find targets in low light conditions and have higher quality scene analysis with sound recognition techniques. In particular, microphone arrays are commonly used since they can estimate sound source directions by calculating the time difference of arrival between microphones. One example of drone audition applications is to carry out people searching tasks [1,2,3]. By estimating the sound direction of people calling for help, drones will be able to find them even if they are covered in rubble. The main goal of this people searching task is to localize a stationary sound source (the person to be rescued), but only localizing stationary sound sources is not enough to accomplish this task. In addition to localization of stationary sources, tracking moving sound sources and sound source separation are also required.

Since different sound sources can exist in the disaster site, drones will also hear non-target sound sources (such as rescue staff, heavy machinery, and so on), and the input to a microphone array is often a mixture of several acoustic signals. In order to enable the drone to recognize which sound source to target, identifying the movement and individual acoustic signal of each sound source is necessary. If the drone realizes that a human-like audio signal has been emitted from a stationary sound source, it is likely to be the target sound source. Besides, research shows that the information of the source location is helpful for sound source separation techniques [4,5], which demonstrates the importance of tracking moving sound sources in drone audition. Therefore, several sound source tracking methods for drones has been proposed [1,6,7,8]. Authors also have proposed a sound source tracking method that utilizes multiple microphone arrays [9,10]. By integrating the spatial spectra computed from each microphone array, the likelihood distribution of the sound source location, which is useful for sound source tracking, can be obtained. Many sound source tracking methods have also been proposed, not only for drones. For example, microphones can be equipped around a room to track human activities [11,12]. Similar to drones, land-based robots can also benefit from microphone arrays, so they can detect and localize sound sources and recognize the environment of the surroundings [13,14,15,16,17]. These tracking methods can be applied to drone systems. However, it is questionable whether these tracking methods are affected by (1) a long distance to the target sound source and (2) significant drone noise. Most research evaluates tracking performance by conducting a few experiments and does not provide an exhaustive evaluation of distance or SNR. For example, Lauzon et al. proposed a sound source tracking method for drone detection and showed its effectiveness through a single outdoor experiment, but tracking performance against various scenarios was not examined [18]. Brandstein et al. also proposed a sound source tracking method and demonstrated experiments to evaluate tracking performance against various source distances. However, the source distance varied between 1 and 3 m, which is not sufficient in terms of drone applications [19]. Since experiments using drones take time and involve high costs, it is difficult to perform exhaustive evaluations through real-world experiments. Therefore, this paper aims to assess the tracking performance of tracking methods by performing numerical simulations for various scenarios, especially various source distances and SNRs.

The rest of the paper is organized as follows. Related work is mentioned in Section 2 and the proposed method [10] is explained in Section 3. The evaluation via numerical simulation is described in Section 4. Finally, the conclusion is written in Section 5.

## 2. Related Work

In the field of acoustic signal processing, microphone arrays are well used to localize the direction or location of the sound source. To track sound source locations using microphone arrays, several measures have been taken. One idea is to estimate the distance between a microphone array and a sound source by applying Bayesian filtering to estimated directions [13,14,20]. However, sound source tracking using a single microphone array is difficult since estimation converging to the ground truth takes time, which is concerning when the sound source is moving.

Therefore, using multiple microphone arrays is much more popular when it comes to tracking sound source locations. As well as using a single microphone array, applying Kalman filtering or particle filtering to estimated directions obtained from multiple microphone arrays shows high tracking performance [18]. Another way is to use triangulation to obtain the sound source location. One of the biggest benefits of using multiple microphone arrays is that we can explicitly obtain the sound source location through triangulation. Since the approximate location can be obtained from one direction estimation, convergence to the ground truth is much faster. Therefore, there are many studies that utilize triangulation for both localizing stationary sound sources [21,22] and tracking moving sound sources [9,23]. If the sound source is assumed to be on the ground, calculating the intersection point between the sound source direction and the ground surface is also a good way to localize the sound source [1,6]. However, when it comes to drone applications, severe ego-noise and wind noise may distort estimation results of the direction, which will negatively affect triangulation methods. Since both commonly used direction estimation and triangulation have discreteness in their calculation, a little distortion will turn to a large tracking error.

Therefore, we introduce a sound source tracking method that estimates the location distribution of the sound source without using triangulation. Generally, direction estimation methods first calculate the likelihood distribution of the direction and then search for the direction with the largest likelihood. In previous work, we integrated likelihood distributions obtained from each microphone array and obtained the location likelihood distribution instead of estimating directions and computing triangulation points [10]. In this way, the information of the sound source location is represented in a location distribution, which is a more continuous representation than a group of triangulation points, and we expect that it will be more robust to drone noise. We apply this location distribution to particle filtering techniques in order to track moving sound sources. In this paper, we will evaluate the tracking performance of existing work [18,23] and the proposed method named particle filtering with integrated MUSIC (PAFIM) [10] via numerical simulation in order to understand the performance against different source distances and different SNRs.

## 3. Method

This section explains PAFIM [10], a sound source tracking method based on location likelihood estimation. The likelihood distribution of the location is obtained by integrating the likelihood distribution of the sound source direction.

### 3.1. Settings

We consider tracking a sound source by multiple microphone arrays. We assume that *N* microphone arrays are mounted to drones and each microphone array is numbered as follows.
$$\begin{array}{c} {\mathrm{MA}}_{1},\ldots ,{\mathrm{MA}}_{n},\ldots ,{\mathrm{MA}}_{N} \end{array}$$

Each microphone array is mounted to a drone, and the state of ${\mathrm{MA}}_{n}$ is described as below, and assumed to be known.
(1)$$\begin{array}{c} m_{n,k}={\left[\begin{array}{cccccc} x_{n,k}, & y_{n,k}, & z_{n,k}, & \phi_{n,k}, & \theta_{n,k}, & \psi_{n,k} \end{array}\right]}^{T} \end{array}$$
where *k* indicates the time step, $x_{n,k}$, $y_{n,k}$, and $z_{n,k}$ indicate the center of ${\mathrm{MA}}_{n}$ in three-dimensional coordinates and $\phi_{n,k}$, $\theta_{n,k}$, and $\psi_{n,k}$ indicate the three-dimensional rotational angles of ${\mathrm{MA}}_{n}$. Each microphone array consists of *M* microphones. We assume the sound source to be a point source, and its location is described as below.
(2)$$\begin{array}{c} e_{k}={\left[\begin{array}{ccc} x_{e,k}, & y_{e,k}, & z_{e,k} \end{array}\right]}^{T} \end{array}$$

The problem addressed in this paper is to estimate the sound source trajectory by repeatedly estimating the sound source location $e_{k}$ from the state of all *N* microphone arrays and recorded sound signal $s_{1},\ldots ,e_{N}\in R^{M}$.

### 3.2. System Outline

This method is based on the integration of direction likelihood distributions obtained by source direction estimation. In most cases, direction estimation methods calculate the likelihood P(ϕ,θ) for each azimuth ϕ and elevation θ, and assume the direction with the largest P(ϕ,θ) is the sound source direction. The approach of this method is to estimate the sound source location by converting the direction likelihood P(ϕ,θ) of each microphone array into the likelihood of a three-dimensional location. Figure 1 illustrates the procedure of the proposed tracking method. This method estimates the sound source trajectory through particle filtering, with each particle being given the location likelihood as its weight.

### 3.3. Estimation of Direction Likelihood Distribution

Some indicators can be regarded as likelihoods for sound source direction. The cross-power spectrum phase (CSP) coefficients are used in the CSP method [24], which is a method for estimating the time difference of arrival (TDOA) with two microphones, and the spatial spectrum obtained from the delay-and-sum beamformer includes scalar quantities whose parameters are the sound source direction, and they generally have a peak in the direction of the source [20]. In this paper, we use a MUSIC spectrum, which produces sharp peaks in the direction of the sound source as the direction likelihood. The MUSIC (multiple signal classification) method analyses the eigenspace of the spatial correlation matrix and estimates the source azimuth and elevation by using the orthogonality between the subspaces of the target sound source and the noise [25]. Let $a(\omega ,\phi ,\theta )\in C^{M}$ be the transfer function of a sound signal of a frequency component ω from a direction (ϕ,θ), then the spatial spectrum, which is known as the MUSIC spectrum, can be expressed as
(3)$$\begin{array}{c} P\left(\phi ,\theta \right)=\frac{1}{\omega_{H}-\omega_{L}+1}\sum_{\omega =\omega_{L}}^{\omega_{H}}\frac{{a(\omega ,\phi ,\theta )}^{H}a(\omega ,\phi ,\theta )}{{a(\omega ,\phi ,\theta )}^{H}E_{N}\left(\omega \right)E_{N}{\left(\omega \right)}^{H}a(\omega ,\phi ,\theta )} \end{array}$$
where $E_{N}$ is a matrix consisting of eigenvectors of the noise subspace of the spatial correlation matrix. In general, the sound source direction is considered to be the direction where the MUSIC spectrum has peaks when estimating the direction. In this paper we regard the MUSIC spectrum as a direction likelihood of the sound source and we integrate the MUSIC spectra obtained from each microphone array and convert them into a location likelihood of the sound source.

### 3.4. Converting to Location Likelihood Distribution

Let $P_{n}\left(\phi ,\theta \right)$ be the MUSIC spectrum calculated from ${\mathrm{MA}}_{n}$. We express the likelihood distribution of the sound location by simply summing $P_{n}\left(\phi ,\theta \right)$. Let an arbitrary three-dimensional location be ***x***, and the direction from point ***x*** to ${\mathrm{MA}}_{n}$ be $\left(\phi_{n},\theta_{n}\right)$. Then, the location likelihood *L* at a location *x* is described as follows.
(4)$$\begin{array}{cc} \; & L\left(x\right)=\sum_{n}P_{n}\left({\tilde{\phi }}_{n}^{\mathrm{round}},{\tilde{\theta }}_{n}^{\mathrm{round}}\right) \end{array}$$
(5)$$\begin{array}{cc} \; & {\tilde{\phi }}_{n}^{\mathrm{round}}=\mathrm{round}\left({\tilde{\phi }}_{n}\right),\;{\tilde{\theta }}_{n}^{\mathrm{round}}=\mathrm{round}\left({\tilde{\theta }}_{n}\right) \end{array}$$
(6)$$\begin{array}{cc} \; & \left[\begin{array}{c} \cos{\tilde{\phi }}_{n}\cos{\tilde{\phi }}_{n}\\ \sin{\tilde{\phi }}_{n}\cos{\tilde{\phi }}_{n}\\ \sin{\tilde{\phi }}_{n} \end{array}\right]=R_{n}^{-1}\left[\begin{array}{c} \cos\phi_{n}\cos\phi_{n}\\ \sin\phi_{n}\cos\phi_{n}\\ \sin\phi_{n} \end{array}\right] \end{array}$$
where round(·) is a function that rounds the direction according to the resolution of transfer function a(ω,ϕ,θ) and $R_{n}$ is the rotation matrix representing the posture of ${\mathrm{MA}}_{n}$, which can be defined by $\left(\phi_{n,k},\theta_{n,k},\psi_{n,k}\right)$. In other words, location likelihood L(x) is the summation of direction likelihoods corresponding to directions towards *x* seen from each microphone array.

### 3.5. Tracking Based on Location Likelihood Distribution

Since we can obtain the location likelihood of an arbitrary 3D location *x*, it is able to track the sound source location by applying this location likelihood L(x) to particle filtering. Let *I* be the number of particles and $x_{k}^{i},w_{k}^{i}$ be, respectively, the state and weight of the *i*-th particle at a time step *k*. The state $x_{k}^{i}$ includes the location and velocity of particle *i*.
(7)$$\begin{array}{c} x_{k}^{i}={\left[\begin{array}{cccccc} x_{k}^{i}, & y_{k}^{i}, & z_{k}^{i}, & {\dot{x}}_{k}^{i}, & {\dot{y}}_{k}^{i}, & {\dot{z}}_{k}^{i} \end{array}\right]}^{T} \end{array}$$

### 3.6. Initialization of Particle Filter

Initial particles are sampled from the following distribution.
(8)$$\begin{array}{cc} x_{0}^{i} & ∼N(\mu_{0},\Sigma_{0}) \end{array}$$
(9)$$\begin{array}{cc} \mu_{0} & ={\left[\begin{array}{cccc} \mu_{0,pos}, & 0, & 0, & 0 \end{array}\right]}^{T} \end{array}$$
(10)$$\begin{array}{cc} \Sigma_{0} & =\left[\begin{array}{cc} \sigma_{pos}^{2}I & O\\ O & \sigma_{vel}^{2}I \end{array}\right] \end{array}$$
where $\mu_{0,pos}$ is the mean of the initial distribution. $\mu_{0,pos}$ can be determined if there is initial information of the sound source, or it can be derived from triangulation based on direction estimation [9]. After calculating triangulation points, $\mu_{0,pos}$ is obtained by taking the average of all triangulation points.

### 3.7. Particle Update

We use an excitation-damping model for the prediction model [18].
(11)$$\begin{array}{cc} \; & x_{k}^{i}=Fx_{k-1}^{i}+Hv \end{array}$$
(12)$$\begin{array}{cc} \; & F=\left[\begin{array}{cc} I & TI\\ O & aI \end{array}\right],\;H=\left[\begin{array}{c} O\\ bI \end{array}\right] \end{array}$$
(13)$$\begin{array}{cc} \; & v∼N(0,I) \end{array}$$
where $I\in R^{3\times 3}$ is an identity matrix and $O\in R^{3\times 3}$ is a zero matrix. Variables *a* and *b* determine the ratio of velocity to carry on from the previous time step and the excitation of particles, respectively. Each particle gains weight proportional to $L(x_{k,pos}^{i})$, hence
(14)$$\begin{array}{c} w_{k}^{i}=w_{k-1}^{i}\frac{L(x_{k,pos}^{i})}{\sum_{i}L\left(x_{k,pos}^{i}\right)} \end{array}$$
where $x_{k,pos}^{i}={\left[x_{k}^{i},y_{k}^{i},z_{k}^{i}\right]}^{T}$. Resampling would be necessary if effective particles are less than a threshold $N_{thr}$. Hence, when
(15)$$\begin{array}{c} \frac{1}{\sum_{i}{\left(w_{k}^{i}\right)}^{2}}\leq N_{thr} \end{array}$$
is satisfied, the weight of each particle should reset to 1/I.

### 3.8. Location Estimation

Finally, the estimated sound source location at time step *k* is obtained by taking the weighted average of the particles.
(16)$$\begin{array}{c} {\hat{x}}_{e,k}=\sum_{i=1}w_{k}^{i}x_{i}^{k} \end{array}$$

## 4. Evaluation

In this section, we perform a numerical simulation to evaluate the tracking performance of several tracking methods. When performing source tracking, various challenges are taken into account such as source distance, SNR, presence of obstacles, microphone array placement, and self-positioning of the drone. However, it is difficult to evaluate all the factors at once because it is difficult to identify which factor is responsible for the performance change. Hence, factors to be evaluated should be narrowed. In outdoor environments the range of the sound source that can be heard can be very wide. Hence, how far the sound source tracking can succeed is a big interest. Besides, since loud drone noise will be input to the microphones, robustness against low SNR is also a big concern. Therefore, in this evaluation, we focus only on the two important factors: source distance and SNR.

### 4.1. Evaluation Outline

We performed a numerical simulation via MATLAB to evaluate the performance of PAFIM and existing methods in sound source tracking. We considered a scenario of sound source tracking with two drones surrounding a single sound source (see Figure 2). Each drone has two microphone arrays equipped as shown in Figure 3 and each microphone array consists of 16 microphones placed spherically (See Figure 4). Microphones record acoustic signals at 16 kHz, 24 bits, and the transfer function used to calculate the MUSIC spectrum has a resolution of 5 degrees. The simulated recording is calculated by a definitive transfer function from the sound source, and noise during the sound transmission is omitted. Both drones are hovering still at a 30 m height, which means they do not move through the simulation.

In this simulation, we focused on the concern in relation to tracking performance with respect to the change in source distance and the change in signal-to-noise ratio (SNR). Therefore, we evaluated the performance of tracking methods by moving the source distance and SNR independently, which means the SNR does not correlate with the source distance. We moved the horizontal distance between the sound source and drones from 10 m to 80 m in increments of 10 m, and we varied the SNR from −60 dB to 10 dB. The range of distance was decided as above since a simulated rescue demonstration has been performed against a sound source 20 to 30 m away [6], and we wanted to evaluate performance limits in terms of distance. The range of SNR was decided by reports of indoor experiments showing that the SNR could be −15 to −25 dB [8,26]. In addition, outdoor environments could make SNR lower, since drones for outdoors tend to make louder noise and wind noise input to the microphones. Drone noise is added to the input by adding prerecorded drone noise to the recorded signal, and the SNR is set by adjusting the amplitude of the drone noise. Drone noise is assumed to be emitted from the red dots illustrated in Figure 3. As the horizontal distance increases, the tracking result is expected to worsen due to the direction estimation’s discreteness. As the SNR decreases, the tracking result is expected to worsen since the direction estimation will be affected by low SNR. In order to evaluate the tracking performance for various source motions, simulations were performed for three types of motion: stationary, circular, and random walk. Stationary motion does not permit the sound source to move at the origin. Circular motion lets the sound source move in a circle with a radius of 5 m and the circle center is set to the origin. While the sound source is moving in a circle, the speed of the sound source is constant, and it takes 10 s to go around the circle. Random walk motion lets the sound source move randomly with the following behavior starting from the origin.
(17)$$\begin{array}{cc} x_{e,k+1} & =x_{e,k}+v_{k+1}\left[\begin{array}{c} \cos\phi_{k+1}\\ \sin\phi_{k+1}\\ 0 \end{array}\right] \end{array}$$
(18)$$\begin{array}{cc} v_{k+1} & =v_{k}+T\Delta v,\;\Delta v∼0.5N\left(0,1\right) \end{array}$$
(19)$$\begin{array}{cc} \phi_{k+1} & =\phi_{k}+T\Delta \phi ,\;\Delta \phi ∼1000\times \frac{\pi }{180}\times N\left(0,1\right) \end{array}$$

For the sound source, we prepared thirty sound clips (ten male voices, ten female voices, and ten white noise clips). Human voice clips were made from a corpus that reads Japanese news [27]. Each tracking method estimated the sound source location every T=0.2 s during 10 s. For all tracking methods, we utilized the MUSIC method for direction estimation (or calculation of location likelihood) in this simulation, although some methods use different direction estimation methods. In the MUSIC method, we used a transfer function with a resolution of 5 degrees, and the range of frequency was $\omega_{L}=200$ Hz to $\omega_{H}=2200$ Hz. Results obtained in this simulation are described in the next subsection.

### 4.2. Simulation Results and Discussion

We mainly evaluate tracking performance by looking at the tracking error. We define the tracking error as the Euclidean distance between the estimated location and the ground truth. We also use root-mean-square error (RMSE), which is defined as below, to evaluate the error of the entire trajectory, where *K* is the number of time steps in the simulation.
(20)$$\begin{array}{c} \mathrm{RMSE}=\sqrt{\frac{1}{K}\sum_{k=1}^{K}{\mathrm{error}}_{k}^{2}} \end{array}$$

Table 1 summarizes the figures showing the tracking errors. Each figure shows the simulated RMSE of each method. Basically, each figure shows the RMSE for each distance and SNR, although for the sake of clarity, figures for a specific distance (=30 m) and for a specific SNR (=−20 dB) are also shown. Given these results, we discuss the tracking performance of the three methods. Through the simulations, significant results were obtained in terms of source distance, SNR, and tracking methods.

**Table 1 ijerph-18-09039-t001:** Table of figures showing RMSE of simulation results for each source movement and sound type.

	Female	Male		White Noise
Stationary	Figure 5	Overall	Figure 6	Figure 7
		Fixed the distance at 30 m	Figure 8	
		Fixed the SNR at −20 dB	Figure 9	
Circle	Figure 10	Overall	Figure 11	Figure 12
		Fixed the distance at 30 m	Figure 13	
		Fixed the SNR at −20 dB	Figure 14	
Random walk	Figure 15	Overall	Figure 16	Figure 17
		Fixed the distance at 30 m	Figure 18	
		Fixed the SNR at −20 dB	Figure 19	

**Figure 5 ijerph-18-09039-f005:**
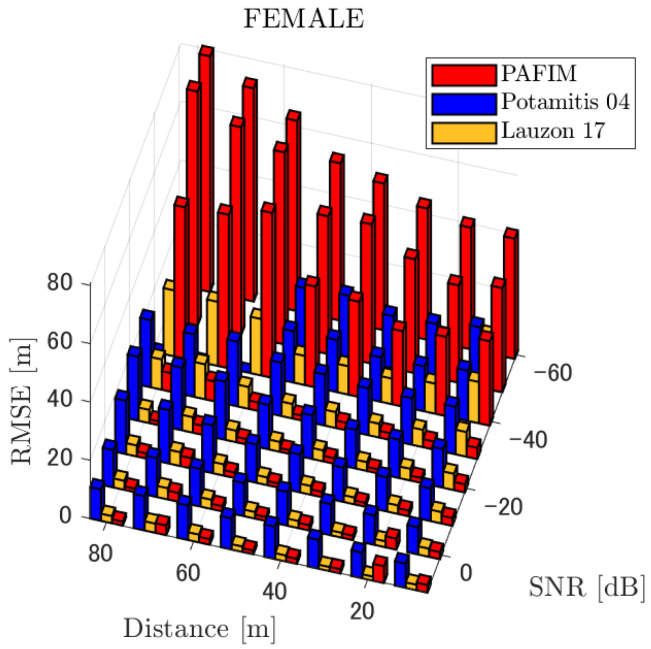
RMSE of tracking a stationary sound source emitting female voice.

**Figure 6 ijerph-18-09039-f006:**
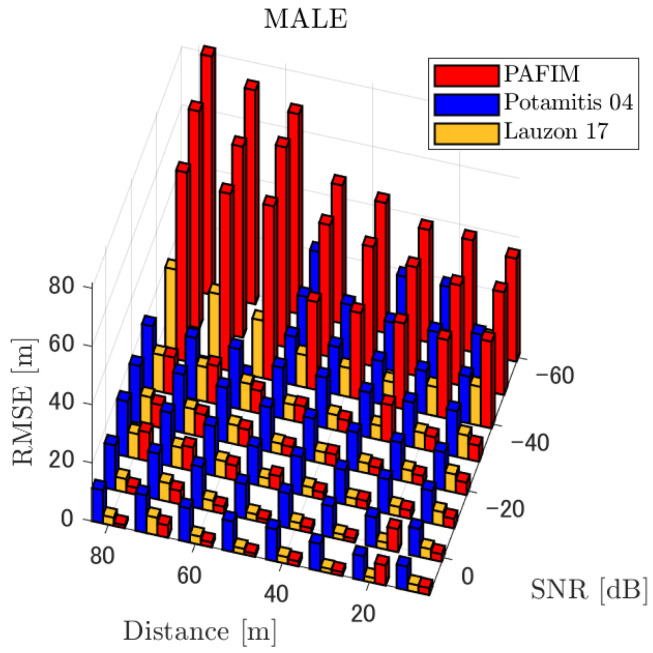
RMSE of tracking a stationary sound source emitting male voice.

**Figure 7 ijerph-18-09039-f007:**
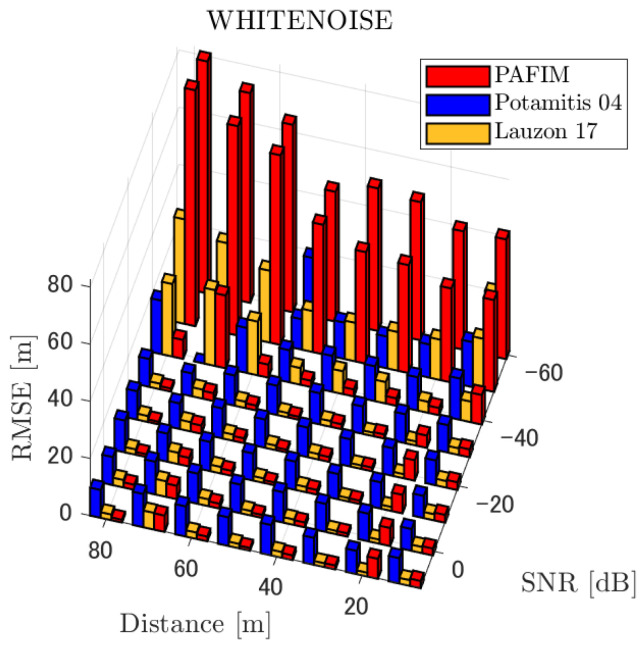
RMSE of tracking a stationary sound source emitting white noise.

**Figure 8 ijerph-18-09039-f008:**
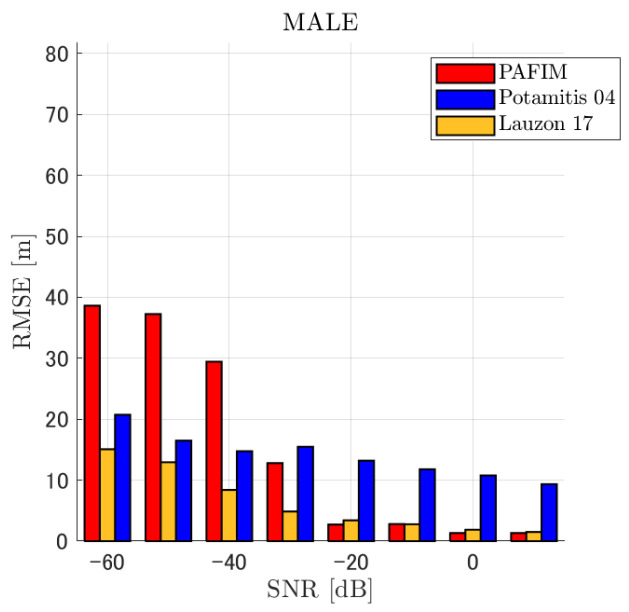
RMSE of tracking a stationary sound source emitting male voice (focusing on source horizontal distance = 30 m).

**Figure 9 ijerph-18-09039-f009:**
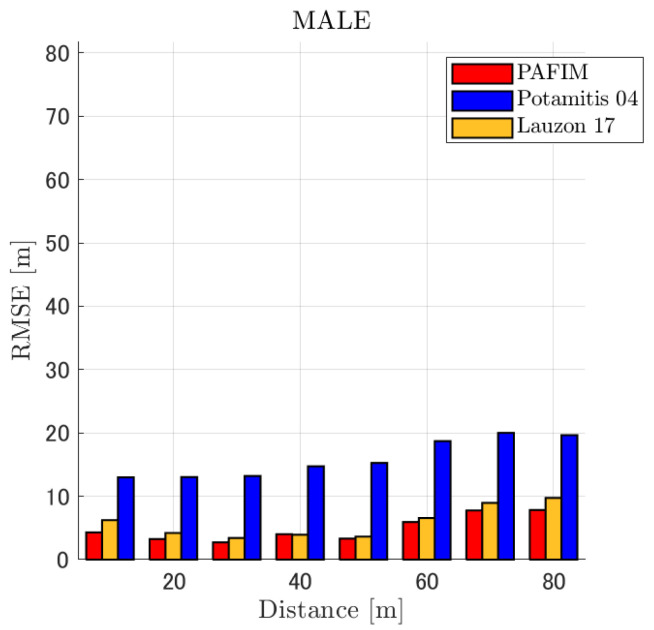
RMSE of tracking a stationary sound source emitting male voice (focusing on SNR = −20 dB).

**Figure 10 ijerph-18-09039-f010:**
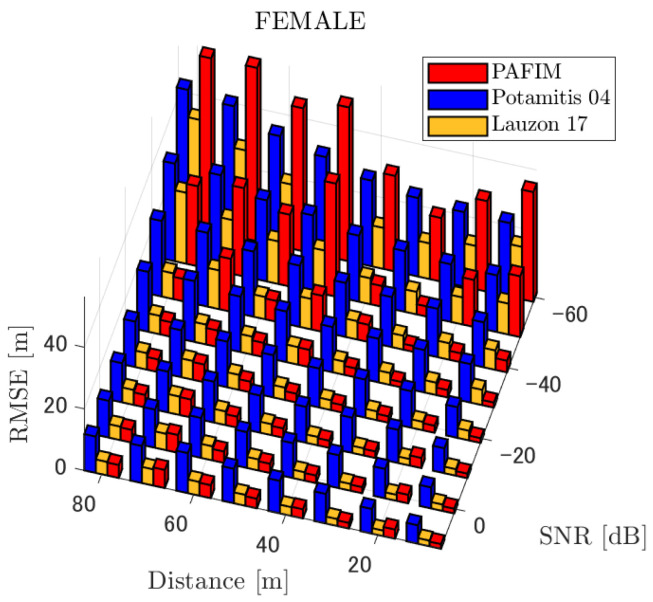
RMSE of tracking a sound source moving in a circle emitting female voice.

**Figure 11 ijerph-18-09039-f011:**
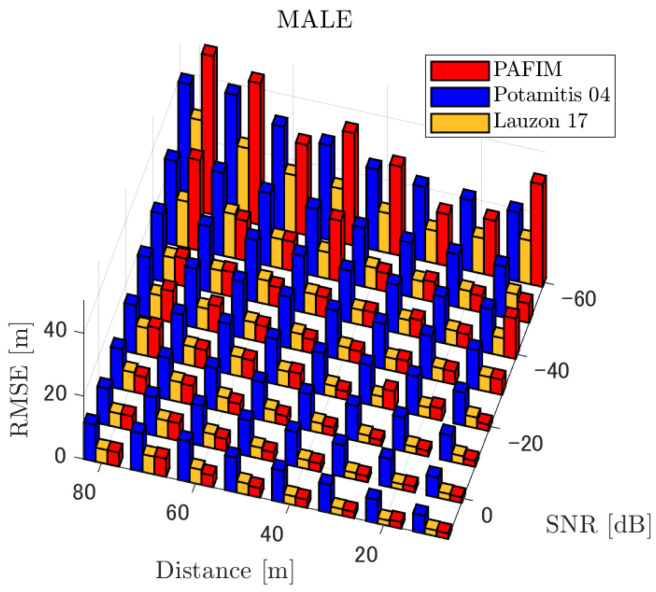
RMSE of tracking a sound source moving in a circle emitting male voice.

**Figure 12 ijerph-18-09039-f012:**
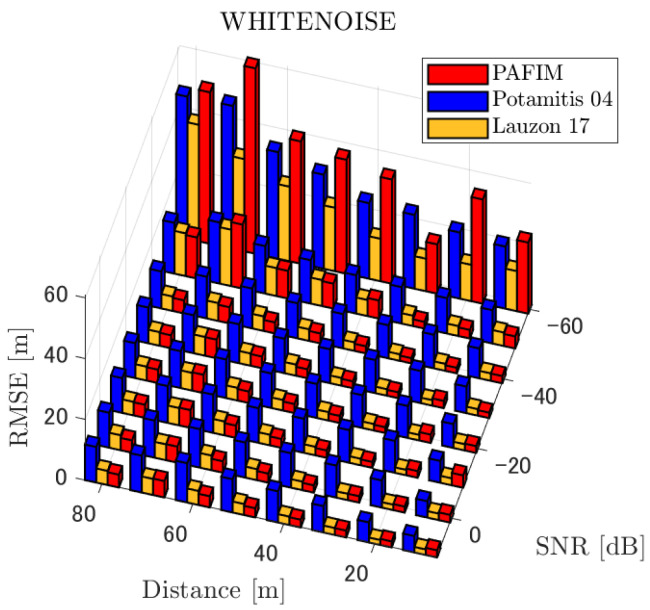
RMSE of tracking a sound source moving in a circle emitting white noise.

**Figure 13 ijerph-18-09039-f013:**
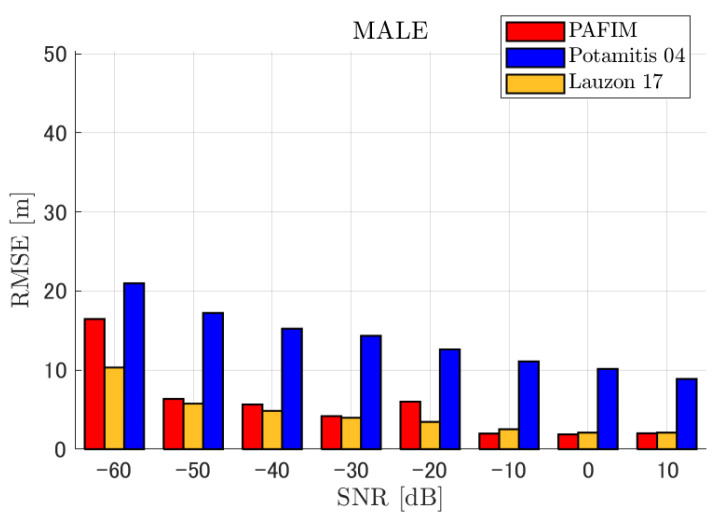
RMSE of tracking a sound source moving in a circle emitting male voice (focusing on source horizontal distance = 30 m).

**Figure 14 ijerph-18-09039-f014:**
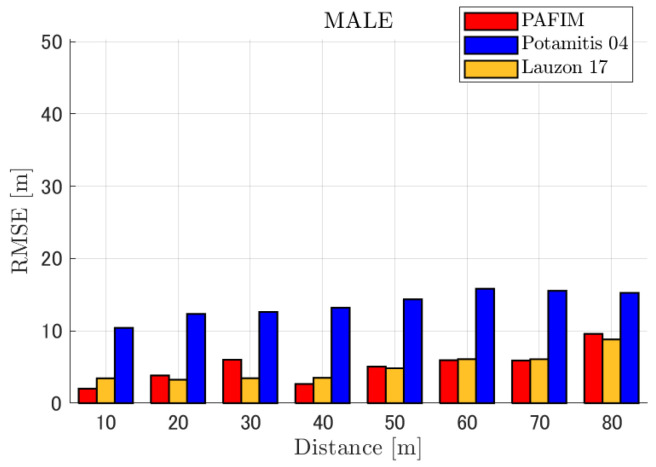
RMSE of tracking a sound source moving in a circle emitting male voice (focusing on SNR = −20 dB).

**Figure 15 ijerph-18-09039-f015:**
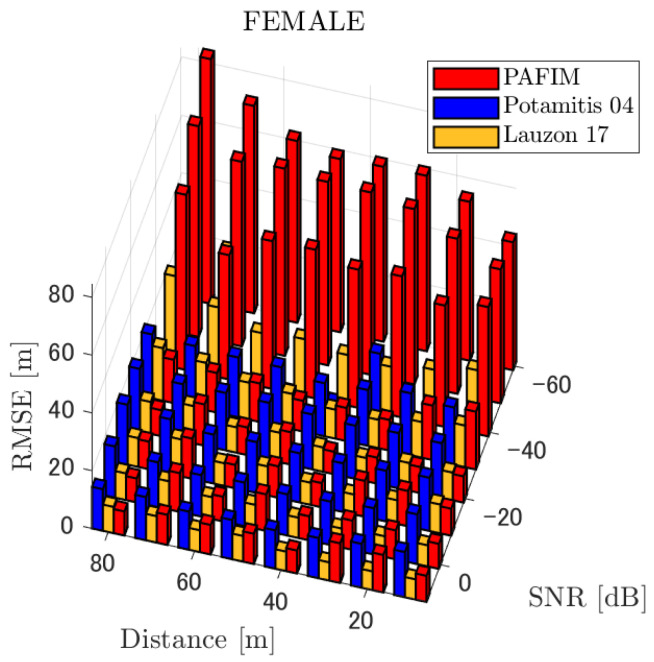
RMSE of tracking a randomly moving sound source emitting female voice.

**Figure 16 ijerph-18-09039-f016:**
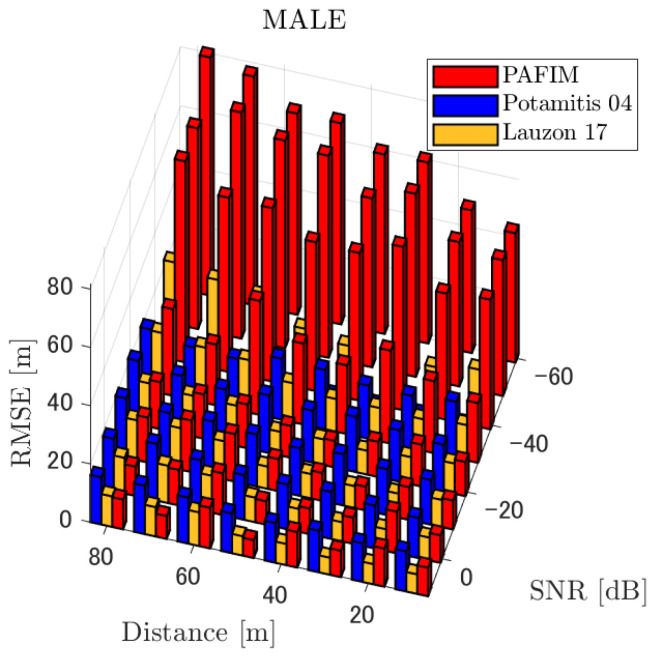
RMSE of tracking a randomly moving sound source emitting male voice.

**Figure 17 ijerph-18-09039-f017:**
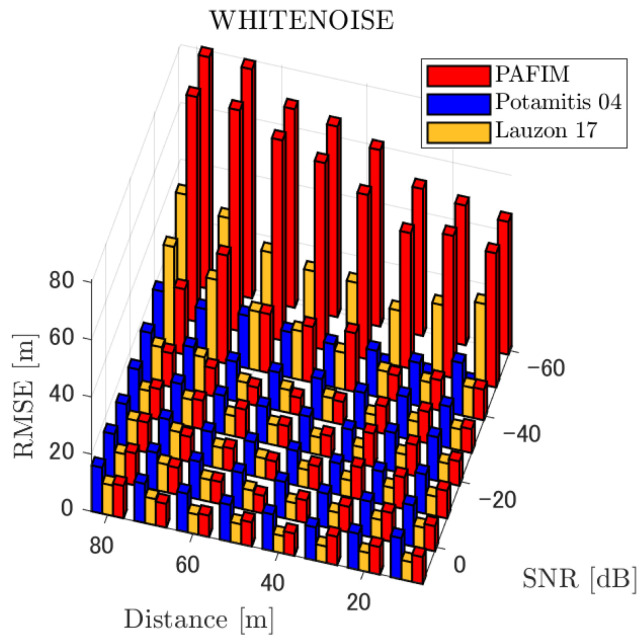
RMSE of tracking a randomly moving sound source emitting white noise.

**Figure 18 ijerph-18-09039-f018:**
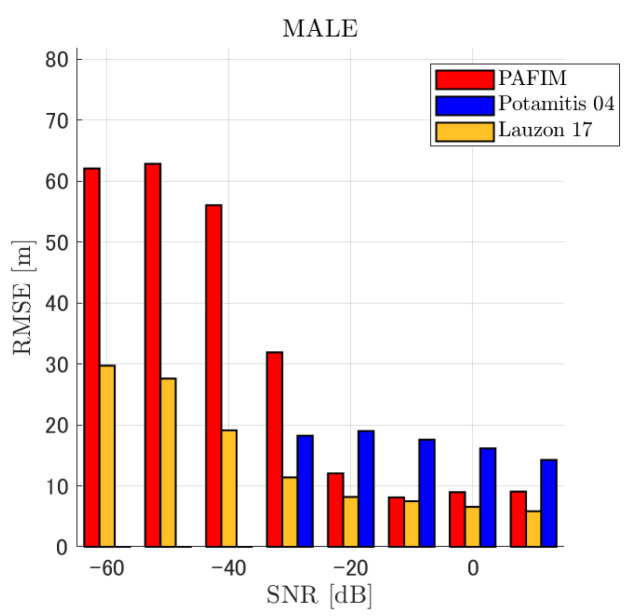
RMSE of tracking a randomly moving sound source emitting male voice (focusing on source horizontal distance = 30 m). Tracking results of Potamitis’s method did not converged when the SNR is less than −40 dB.

**Figure 19 ijerph-18-09039-f019:**
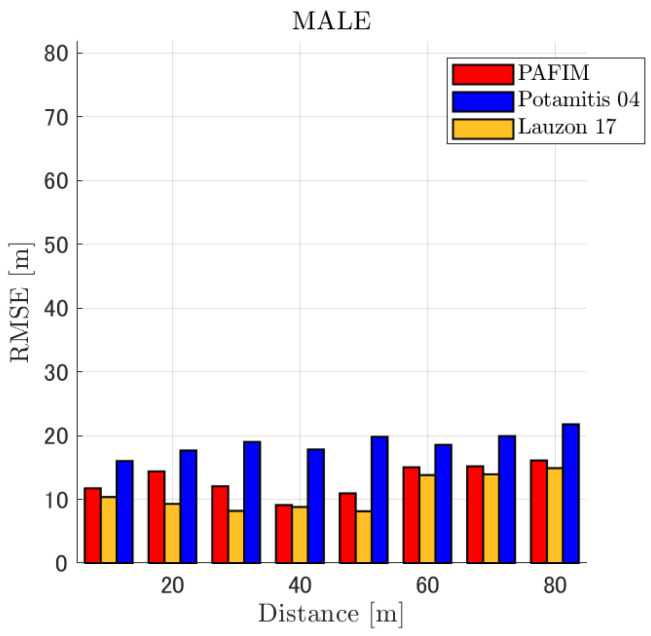
RMSE of tracking a randomly moving sound source emitting male voice (focusing on SNR = −20 dB).

#### 4.2.1. Discussion: Distance

For any source movement, the tracking error increases as the source distance increases. We believe that the main cause of this phenomenon is the resolution of the transfer function a(ϕ,θ), which is used in every method. As in the simulation, if the resolution is five degrees, the estimation error at a point *l* m away will be πl/36 m if the source direction estimation is off by one element. For example, if the sound source is 50 m away, the source location estimation error will be about 4.4 m. This tendency can be seen in the RMSE of PAFIM from Figure 9 and Figure 14.

#### 4.2.2. Discussion: SNR

Similar to distance, tracking error increases as the SNR decreases. Besides, this increase is more noticeable than the increase due to distance. This is because the direction estimation error is largely affected by SNR rather than the source distance, which could be seen in Figure 20 and Figure 21. Both figures illustrate the direction estimation error of one microphone array by box plot. The direction estimation error significantly deteriorates when the SNR gets lower than −40 dB. This can be taken as a threshold to determine the level of effectiveness of the source direction estimation. When the SNR surpasses this threshold, the RMSE of each method rises sharply. In this simulation, we used the SEVD-MUSIC method, which is the simplest MUSIC method that does not use noise reduction algorithms. Therefore, SEVD-MUSIC has difficulty outputting a proper MUSIC spectrum when the SNR is lower than 0 dB. In this simulation, drone noise is set to be output from the red dots in Figure 3 while the target sound source emits from a single point. It seems that even if the SNR is lower than 0 dB since the drone noise emits from several points, the directionality has blurred compared to the target signal, which has lowered the threshold SNR.

#### 4.2.3. Discussion: Overall Performance

In this section, we assess the overall performance by comparing the simulated methods. Since the nature of tracking error differs between the change in distance and SNR, the nature of the source tracking error for each tracking method also differs. When the source distance is large, the tracking errors of PAFIM [10] and method [18] do not get larger compared to method [23]. This is considered because the source location is represented in the form of a location distribution rather than a discrete representation such as triangulation points or source directions. The discreteness of triangulation points leads to large tracking error because even if the direction estimation error is minimum (=5 degrees) the location error of the triangulation points will be about 3 to 5 m and this gets worse as the source distance increases. However, PAFIM reduces the increase in tracking error with increasing distance by using all the microphone arrays to compute a location distribution. Comparing PAFIM and method [18], PAFIM has performed slightly better tracking against increasing distance since PAFIM utilizes the entire MUSIC spectrum (rather continuous information) while method [18] only uses the direction that takes the peak of the MUSIC spectrum (rather discrete information).

In terms of SNR, we see that PAFIM has a larger error than other methods when the SNR gets lower, and method [23], which uses triangulation, sometimes results in uncomputable error since direction estimation fails and triangulation cannot be performed. Two main reasons are considered as to why PAFIM fails in sound source tracking. Since PAFIM tries to reduce discreteness by estimating the source location distribution, a non-informative distribution will be obtained in a situation where direction estimation fails repeatedly, which means the likelihood distribution of the source direction is distorted. Regardless of the iterative estimation, if the source distribution is highly skewed, it will not converge to the correct distribution. However, methods with triangulation or direction estimation have a chance to capture the correct location or direction of the sound source at some time step, which can lead the tracking to the correct trajectory. This might be one of the reasons why method [18,23] has a smaller RMSE than PAFIM. Another reason for PAFIM’s vulnerability to SNR is that PAFIM does not have a reliable initialization method and has to rely on triangulation in the first step. As mentioned before, triangulation is not reliable when the SNR is too low, and it is difficult for particle filters to get to the correct trajectory when the initial state is too far from the ground truth. However, PAFIM only shows the worst results when the SNR is lower than the assumptions of the direction estimation method. If the direction estimation method can withstand the SNR, we see that PAFIM can outperform other methods from Figure 8, Figure 13 and Figure 18.

In short, numerical simulations show that PAFIM is strong against distant sound sources compared to existing methods since PAFIM tries to reduce the discreteness of direction estimation. Since drones generally perform their tasks in vast fields, this is a key strength for drone audition. When it comes to low SNR, PAFIM has the largest tracking error since it is hardly possible to obtain a feasible location distribution. However, SNR observed in real-world experiments is about −20 dB, for which PAFIM had the least tracking error. In summary, PAFIM was found to be more robust to increasing distance than the compared methods if the drone noise was not less than −20 dB.

## 5. Conclusions

In this paper, we evaluated the performance of sound source tracking among several tracking methods that can be used in drone audition. In the numerical simulation, we focused on the performance difference against source distance and SNR. Simulation results address that the nature of tracking error is different between changes in distance and SNR, and this difference brings to light the advantages and disadvantages of tracking methods. The proposed method (PAFIM), which integrates MUSIC spectra to capture the sound source location distribution, has been found to be robust when the target sound source is far from the drones. However, methods that utilize discrete information such as directions and triangulation points are sensitive to low SNR since there is a chance to obtain the correct direction even if the MUSIC spectrum is noisy, while PAFIM fails to capture the location distribution with low SNR. However, when the SNR is about −20 dB or more, PAFIM has still shown the lowest RMSE. In this paper, the source distance and SNR were changed independently, but generally, the SNR increases if the drone gets nearer to the sound source. Therefore, PAFIM can be improved if the drones can be maneuvered with proper action planning that can get close enough to target sound sources and cover unobserved areas. Based on this concept, assessment of how placement of microphone arrays will affect the tracking and action planning for drones is one of the issues to be addressed in the future. In addition, drone noise reduction methods were not applied in the simulations. Due to the fact that PAFIM does not involve conventional source direction estimation, the benefit of denoising methods to PAFIM is unclear. It is also a future task to evaluate how much the PAFIM is improved by noise removal by comparing it with other methods.

## Figures and Tables

**Figure 1 ijerph-18-09039-f001:**
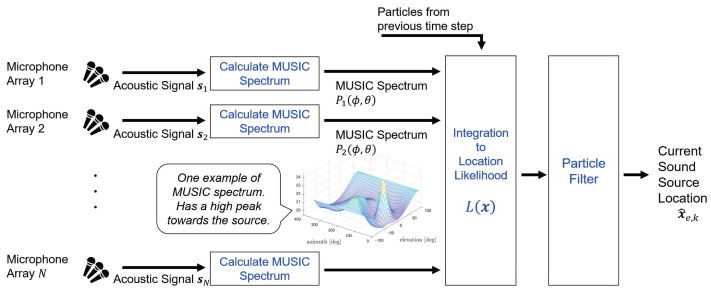
Procedure of proposed method (for one time step).

**Figure 2 ijerph-18-09039-f002:**
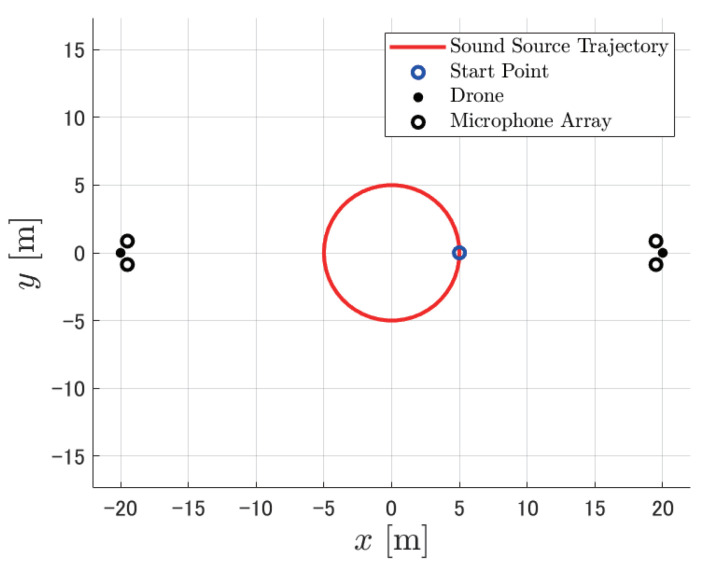
Top view of simulation scenario (horizontal distance = 30 m, sound source movement = circle).

**Figure 3 ijerph-18-09039-f003:**
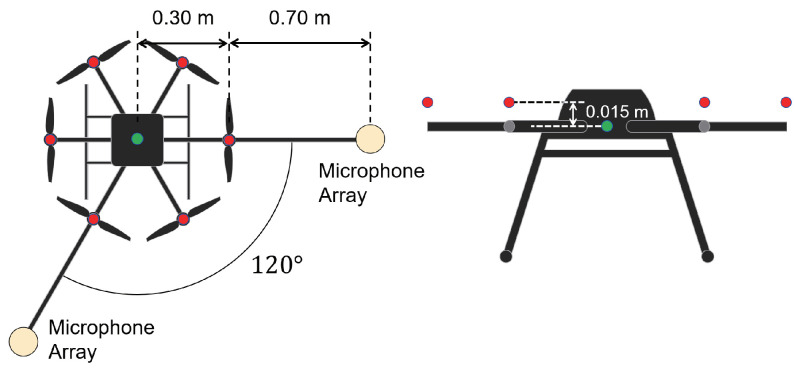
Drone noise source (red dots) in the simulation.

**Figure 4 ijerph-18-09039-f004:**
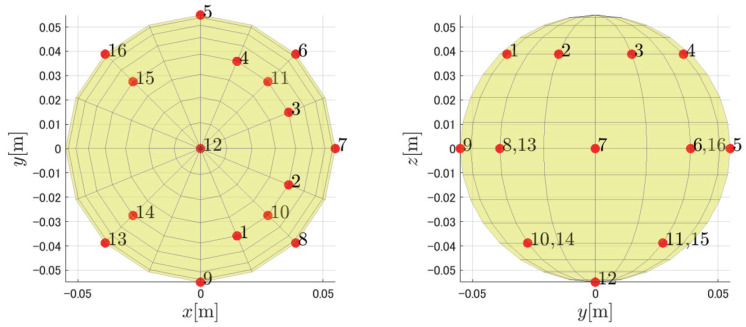
Microphone placement in a microphone array.

**Figure 20 ijerph-18-09039-f020:**
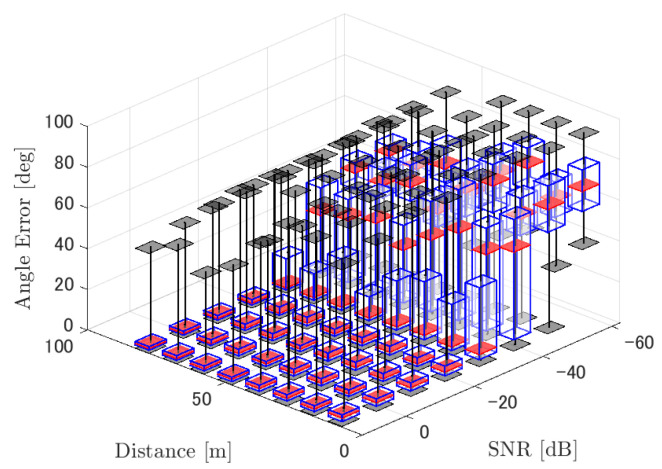
Box plot of the direction estimation error for the lower right microphone array in Figure 2. Considered scenario is tracking a circular motion sound source emitting a male voice.

**Figure 21 ijerph-18-09039-f021:**
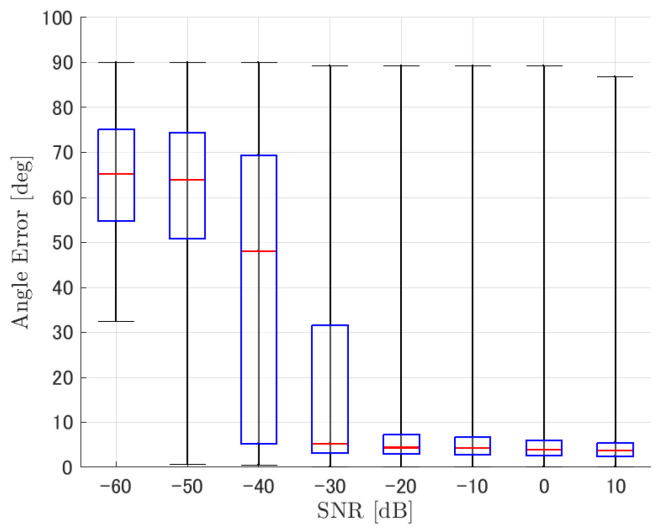
Direction estimation error shown in Figure 20 focusing on distance = 30 m.

## Data Availability

Restrictions apply to the availability of these data. Data was obtained from The Acoustical Society of Japan (2006): ASJ Japanese Newspaper Article Sentences Read Speech Corpus (JNAS) and are available at https://doi.org/10.32130/src.JNAS with the permission of The Acoustical Society of Japan.

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
