# Peer review of "Assessment of Sound Source Tracking Using Multiple Drones Equipped with Multiple Microphone Arrays"

_ijerph, 2021, doi:10.3390/ijerph18179039_

Round 1

Reviewer 1 Report

The authors numerically studied the influence of source distances and SNRs on the evaluation accuracy of several drone-based sound source tracking methods. The study is important for the actual application of drone audition in disaster rescue. However, presentation of the results should be improved. For example, Figs. 5-19 are too boring to read, while the explanations of results are started in the next section! The other related issues are listed below:

1 Abbreviation of a specific technique word can only be used after giving the exact definition. e.g. Line 88, PAFIM is used but the definition is given in the next section.

2 Sound source is given, how to derive the signals captured at microphone arrays equipped with the drones? If the data is obtained by a definitive transfer function from the source, the noise in the sound transmission route is omitted, which makes the simulation far from the real application. The importance of the conclusions is lowered.

3 The authors attribute the increase of tracking error with the increasing source distance to the resolution of the transfer function. Since every method employs this transfer function, why is the influence different?

4 Are the distribution of microphones in the array and the site of array related with the effects of source distance and SNR?

5 The conclusions are weak. Is any improvement of the method possible according to the analysis, which could strengthen the technical sound of this study?    

Author Response

Dear Reviewer 1

Thank you very much for reviewing our manuscript and offering valuable advice.
We have addressed the following comment and revised the manuscript accordingly.

[0] "However, presentation of the results should be improved. For example, Figs. 5-19 are too boring to read, while the explanations of results are started in the next section!"
-> We appreciate your suggestion very much. We have combined the Result section and Discussion section in order not to feel the gap between these two. We also have added a table that organizes the figures which helps the readers to understand what the figures mean at a glance.

[1] "Abbreviation of a specific technique word can only be used after giving the exact definition. e.g. Line 88, PAFIM is used but the definition is given in the next section."
-> Thank you very much for pointing out the error in our sentence structure. We have added the definition of the abbreviation PAFIM where it appears first (line 90).

[2] "Sound source is given, how to derive the signals captured at microphone arrays equipped with the drones? If the data is obtained by a definitive transfer function from the source, the noise in the sound transmission route is omitted, which makes the simulation far from the real application. The importance of the conclusions is lowered."
-> We apologize for our insufficient explanation of the recording environments and the objective of our simulations. The data is obtained by a definitive transfer function, so it is true that the noise in the sound transmission route is omitted. However, this paper focuses on the change in tracking performance for varying distance and SNR. We are aware that in addition to distance and SNR performance, there are other challenges in drone tracking, such as drone self-positioning error, microphone array placement, and of course noise during transmission too. Since it is difficult to evaluate all the factors at once, we think evaluating them one by one is important to unravel the difficulty of sound source tracking. This is one of the reasons why we use numerical simulations instead of experiments in a real environment. For this reason, we decided not to take propagation noise into account. In this simulation, the main parameters, distance and SNR, were changed independently, and we were able to see the change in performance of each method for each change, so we believe that the importance of the conclusion is not small. In response to the remarks, we added information that route transmission noise is omitted (line 146) and we added a sentence emphasizing that the main purpose of this simulation is to consider only the effects of distance and SNR (starting from line 130).

[3] "The authors attribute the increase of tracking error with the increasing source distance to the resolution of the transfer function. Since every method employs this transfer function, why is the influence different?"
-> We are sorry that the consideration against the results was weak. We think the main reason is that what each method mainly estimates is the source tracking. The proposed method PAFIM is based on the integration of MUSIC spectra, Potamitis's method is based on the triangulation points and Lauzon's method is based on the estimated direction of each microphone array. The method using triangulation points is vulnerable to the discreteness of the transfer function, and the location of triangulation points changes significantly when the direction estimation is off by 5 degrees. This gets worse when the distance is large. In addition, triangulation points are usually obtained from two microphone arrays, not all arrays, and even if the triangulation points are obtained from all pairs of microphone arrays, they do not seem to be complementary enough. On the other hand, PAFIM and Lauzon's method integrates the information obtained from all the microphone arrays, which reduces the error caused by discreteness.
The difference between PAFIM and Lauzon's methods is that PAFIM uses the entire MUSIC spectrum, which can be regarded as a two-dimensional array, while Lauzon uses only the direction of the peak. In the case of using only the estimated direction (which is Lauzon's approach), the discrete nature of the transfer function can lead to large location estimation errors when the source distance is long. However, when the SNR is small and the MUSIC spectrum is jaggy, the correct source direction can be obtained occasionally, and the tracking performance can be improved.
Thanks to your questions we have added this discussion in the Result and Discussion section to provide readers further discussion (line 189 to 193).

[4] "Are the distribution of microphones in the array and the site of array related with the effects of source distance and SNR?"
-> Thank you for presenting an interesting challenge for this research. We believe that there is a relationship between the two, although we cannot say for sure because we have not examined it in this paper. In particular, the relationship between the arrangement of the microphone array and the source distance seems to be significant. In this simulation, the two drones are placed around the sound source, but of course, there are other ways to place them. However, as mentioned in [2], it is difficult to verify many factors at once, so we did not consider the microphone array placement in this study. However, it is a very interesting point for our future work and we are very eager to figure out what microphone array placements can do to tracking performances. We have added this point in the Conclusion section as one of the future works (From the end of line 233 to 237).

[5] The conclusions are weak. Is any improvement of the method possible according to the analysis, which could strengthen the technical sound of this study?  
-> We apologize that the conclusions are unclear and weak, and we appreciate you pointing this out. Since we found that PAFIM has a weakness against too low SNR but is yet useful in normal conditions, we think that the mobility of drones can help the task of tracking. Generally, SNR gets large when we can get near the target sound source. So we can form an action planning problem for drones that can improve the performance of PAFIM. In this sense, the placement of microphone arrays will be important, and comments mentioned in [4] should also be discussed in the future. In response to the remarks, we added further discussion about the future work as mentioned above (From line 232 to 240).

Sincerely,
Taiki Yamada

Reviewer 2 Report

  • Abstract could have included the results achieved as a result of this study. Well defined results could elaborate the key findings of this paper.
  • Please look at the word “PArticle” at line 92.
  • The location matrix seems rather 2-D. the 3rd variable seems to be derived from the two.
  • Results obtained are useful and show some conformance.
  • Please proofread the manuscript before final submission.

Author Response

Dear Reviewer 2

Thank you very much for reviewing our manuscript and offering valuable advice.
We have addressed the following comment and revised the manuscript accordingly.

[1] "Abstract could have included the results achieved as a result of this study. Well defined results could elaborate the key findings of this paper."
-> Thank you for your recommendation. We have added a further explanation of the results (mentioned in the Discussion section) at the end of the abstract (line 13).

[2] "Please look at the word “PArticle” at line 92."
-> The letter "A" was capital intentionally to clarify what the abbreviation "PAFIM" stands for. However, we agree it is incorrect English so we have fixed it to lower case (line 90).

[3] "The location matrix seems rather 2-D. the 3rd variable seems to be derived from the two."
-> We're afraid to say that we didn't get what "the location matrix" refers to. We would like to remove the incomprehension in our equations, so could you tell us which line of the equation seems rather 2-D?

[4] "Please proofread the manuscript before final submission."
-> Thank you very much for your kind recommendation. However, we have to apologize that we cannot proofread during this revision since it is a long holiday in the author's area. We guarantee that the final submission will be proofread.

Sincerely,
Taiki Yamada

Reviewer 3 Report

Actual Problem and well described aim, research procedure and results analisys.

Good initial research work in this area of problems. The conlusion: "In conclusion, we can see from the numerical simulation that PAFIM is a suitable sound source tracking method for drones." is too strong in my opinion. Some additional research is necessary.

References are modern and adequate. There are also many other research works you could refer.

I have accepted the paper in the present form.
My comment were to the authors for their future work.
My statement: "The conlusion: 'In conclusion, we can see from the numerical simulation that PAFIM is a suitable sound source tracking method for drones.' is too strong in my opinion. Some additional research is necessary." means that this is valid only regarding the discussed methods. Adding this, the paper sound will be more professional.
There are thousands of papers, appropriate to be referred here. I accept the author's choice.
All the remarks I made because I would like to be of help to the authors in future.
Finally, I accept the paper for publishing in the presented form. 

Author Response

Dear Reviewer 3

Thank you very much for reviewing our manuscript and offering valuable advice.
We have addressed the following comment and revised the manuscript accordingly.

'In conclusion, we can see from the numerical simulation that PAFIM is a suitable sound source tracking method for drones.' is too strong in my opinion."
-> We agree that this statement is too much to say from the given simulation results. We have softened the words to 'In summary, PAFIM was found to be more robust than compared methods to increasing distance if the drone noise is not smaller than -20 dB.' which is more objective (line 219).

Sincerely,
Taiki Yamada

Reviewer 4 Report

I have no objections to the paper. The work has great utilitarian values.

In this paper, the autors have evaluated the performance of sound source tracking  among several tracking methods that can be used in drone auditions. In numerical simulation, they analyzed the performance of individual tracking systems depending on the source distance and SNR. The simulation results indicate that the nature of the tracking error is more SNR dependent than distance dependent. The advantages and disadvantages of selected tracking methods are also given. The authors also provided future directions for the development of a universal sound tracking system. The work has great utilitarian values and can be widely used in both civil and military work.

Author Response

Dear Reviewer 4

Thank you very much for reviewing our manuscript and giving us kind comments.
We believe that the comments from reviewers have made this manuscript even better.

Sincerely,
Taiki Yamada

Round 2

Reviewer 1 Report

I do not have any additional comments and suggestion. Polishing the language is required before publication.